# Polar localization of CheO under hypoxia promotes *Campylobacter jejuni* chemotactic behavior within host

Ran Mo[1,2,3,4], Wenhui Ma[5], Weijie Zhou[5], Beile Gao[1,2,3]*

**1** CAS Key Laboratory of Tropical Marine Bio Resources and Ecology, Guangdong Key Laboratory of Marine Materia Medica, Innovation Academy of South China Sea Ecology and Environmental Engineering, South China Sea Institute of Oceanology, Chinese Academy of Sciences, Guangzhou, China, **2** Southern Marine Science and Engineering Guangdong Laboratory (Guangzhou), Guangzhou, China, **3** Tropical Marine Biological Research Station in Hainan, Sanya Institute of Oceanology, Chinese Academy of Sciences and Hainan Key Laboratory of Tropical Marine Biotechnology, Sanya, China, **4** University of Chinese Academy of Sciences, Beijing, China, **5** Department of General Surgery & Guangdong Provincial Key Laboratory of Precision Medicine for Gastrointestinal Tumor, Nanfang Hospital, First Clinical Medical School, Southern Medical University, Guangzhou, China

* gaob@scsio.ac.cn

**Data Availability Statement:** All relevant data are within the manuscript and its Supporting information files.

**Funding:** This research was supported by National Natural Science Foundation of China (31870064) to

## Abstract

*Campylobacter jejuni* is a food-borne zoonotic pathogen of worldwide concern and the leading cause of bacterial diarrheal disease. In contrast to other enteric pathogens, *C. jejuni* has strict growth and nutritional requirements but lacks many virulence factors that have evolved for pathogenesis or interactions with the host. It is unclear how this bacterium has adapted to an enteric lifestyle. Here, we discovered that the CheO protein (CJJ81176_1265) is required for *C. jejuni* colonization of mice gut through its role in chemotactic control of flagellar rotation in oxygen-limiting environments. CheO interacts with the chemotaxis signaling proteins CheA and CheZ, and also with the flagellar rotor components FliM and FliY. Under microaerobic conditions, CheO localizes at the cellular poles where the chemosensory array and flagellar machinery are located in *C. jejuni* and its polar localization depends on chemosensory array formation. Several chemoreceptors that mediate energy taxis coordinately determine the bipolar distribution of CheO. Suppressor screening for a Δ*cheO* mutant identified that a single residue variation in FliM can alleviate the phenotype caused by the absence of CheO, confirming its regulatory role in the flagellar rotor switch. CheO homologs are only found in species of the *Campylobacterota* phylum, mostly species of host-associated genera *Campylobacter*, *Helicobacter* and *Wolinella*. The CheO results provide insights into the complexity of chemotaxis signal transduction in *C. jejuni* and closely related species. Importantly, the recruitment of CheO into chemosensory array to promote chemotactic behavior under hypoxia represents a new adaptation strategy of *C. jejuni* to human and animal intestines.

BG, Key Special Project for Introduced Talents Team of Southern Marine Science and Engineering Guangdong Laboratory (Guangzhou) (GML2019ZD0407) to BG, Strategic Priority Research Program of the Chinese Academy of Sciences (XDA19060301) to BG, and Innovation Academy of South China Sea Ecology and Environmental Engineering, Chinese Academy of Sciences (NO. ISEE2021ZD03, ISEE2021PY05) to BG. The funders had no role in study design, data collection and analysis, decision to publish, or preparation of the manuscript.

**Competing interests:** The authors have declared that no competing interests exist.

## Author summary

Bacteria use chemotaxis to navigate their flagellar motility towards or away from a variety of environmental stimuli. For many pathogens, chemotactic motility plays an important role in infection and disease. Understanding the mechanism of chemotaxis behavior in pathogens can help the development of therapeutic strategies by interfering with chemotactic signal transduction. In this study, we identified a novel chemotaxis protein CheO in *Campylobacter jejuni*, a leading cause of human gastroenteritis worldwide. We demonstrated that CheO is directly involved in chemotactic control of the flagellar motor switch, the reason that it is required for colonization of different animal models. We also provide evidences that CheO is responsive to environmental oxygen variation, with a more prominent role in energy taxis under low oxygen levels. Therefore, CheO presents a novel mechanism for *C. jejuni* adaptation to hypoxia conditions such as those existing in human and animal intestines. Targeting CheO and other chemotaxis regulators could reduce the survival of *C. jejuni* within hosts and in the food chain.

## Introduction

*Campylobacter jejuni*, a zoonotic pathogen, is the leading cause of human bacterial enteritis and a primary etiological agent for ruminant abortion [1–4]. It also establishes commensalism with many wild and agriculture-associated animals [5]. For example, *C. jejuni* can quickly reach very high numbers in chicken intestinal tracts but not cause disease. This commensalism with poultry leads to food chain contamination and is a main source of human infection [6–8]. Understanding how *C. jejuni* adapts to its enteric lifestyle is essential to its epidemiology and the development of control strategies.

Compared to other common food-borne enteric pathogens such as *Escherichia coli* and *Salmonella* spp., *C. jejuni* has a streamlined genome of ~1.6Mb and lacks classical virulence factors such as Type III secretion system, Type IV pilus, and various toxins [9,10]. Another unique and also paradoxical feature of *C. jejuni* is its tolerance to oxygen variation. This bacterium is microaerophilic, requiring only a limited amount of oxygen (5%-10%) to grow but able to survive high oxygen tensions during transmission in the food chain [11,12]. Previous studies demonstrated that *C. jejuni* has evolved metabolic pathways and a complex branched electron transport chain to support microaerobic growth [13–16]. In addition, *C. jejuni* possesses a repertoire of oxidative stress defense enzymes to detoxify reactive oxygen species and also produce hemerythrins to protect essential metabolic enzymes during oxygen exposure [17–19]. Other factors beyond metabolism and respiration also contribute to microaerobic adaptation. For example, the percentage of lysophospholipids in the membrane phospholipid of *C. jejuni* is responsive to oxygen levels, and lysophospholipids are required for its normal motility at low oxygen availability [20].

Chemotaxis and flagellar motility play important roles in the establishment and adaptation of *C. jejuni* within its hosts [21,22]. Their crucial roles were highlighted repeatedly by several comprehensive genome-wide analyses to identify fitness determinants for animal model infection using multiple *C. jejuni* strains [23–26]. Among the fitness gene lists required for colonization of chicken, piglet, or mouse models, approximately 15%-24% genes are involved in chemotaxis and flagellar motility [24–26]. Therefore, many studies have been conducted to characterize the chemotaxis system of *C. jejuni*, which displays remarkable differences from the *E. coli* paradigm [27,28]. The chemotaxis system in *E. coli* is composed of five chemoreceptors to detect chemical concentration changes and three core signaling proteins including

CheW, CheA, and CheY that transduce the signal to the flagellar motor [29]. In addition, adaptation system CheBR and phosphatase CheZ can tune the chemotaxis signal amplitude. Compared to this chemotaxis model, *C. jejuni* has additional components to conduct signal sensing and transduction. These components include: approximately 10 chemoreceptors (also called **t**ransducer-**l**ike **p**roteins, Tlps), double the sensory repertoire of *E. coli*; a two-protein pair CheP and CheQ dedicated for transcriptional regulation of the core *cheVAW* operon; coupling protein CheV in addition to CheW; a newly identified signaling protein ChePep that is restricted to the *Campylobacerota* phylum, previous ε-proteobacteria class [27,30–33].

There is a steep oxygen gradient across the host mucus layer where *C. jejuni* predominantly resides; thus, a chemotactic behavior to meet its low but essential oxygen needs is important for successful colonization and pathogenesis [34]. However, it is unclear how the chemotaxis system of *C. jejuni* senses and responds to environmental oxygen variation. For the chemosensory repertoire of *C. jejuni*: Tlp1, 2, 3, 4, 7, 10 and the recently characterized Tlp11 encode periplasmic sensory domains that sense various external ligands; Tlp9 (CetA) in complex with CetBC and Tlp8 (CetZ) measure internal metabolic status and control energy taxis; Tlp6, a homolog of TlpD in *Helicobacter pylori*, may mediate tactic response to oxidative stress through its C-terminal zinc-binding domain (CZB domain) (S1 Fig) [35–41]. Currently, no Tlps in *C. jejuni* have been reported to be oxygen sensors, and it is unclear whether other chemotaxis components function differently under oxygen fluctuation.

Here, we report a previously uncharacterized protein CJJ81176_1265 in *C. jejuni* 81–176 that functions as a chemotaxis signaling protein. This protein can interact with chemotaxis proteins and target the flagellar motor switch proteins to regulate motility, independent of CheY. Interestingly, its polar localization is responsive to the environmental oxygen level and dependent on the chemosensory array formation and Tlps involved in energy taxis. Therefore, we named CJJ81176_1265 as CheO (**Che**motaxis protein responsive to **O**xygen). CheO is required for colonization of the animal gut and its homologs are commonly found in species of *Campylobacter*, *Helicobacter*, and *Wolinella* that are host-associated commensals or pathogens. Our results suggest that CheO is a chemotaxis protein that can promote *C. jejuni* colonization and adaptation to host niches with low oxygen concentrations.

## Results

### CheO confers a fitness advantage for *C. jejuni* colonization of animal models by affecting chemotaxis behavior

TnSeq screenings of *C. jejuni* mutant libraries in animal models identified 15%-25% genes of unknown function but worth investigation due to the strong colonization defects of their mutants [24–26,42]. One of these genes was CJJ81176_1265 (*cheO*) as a potential fitness determinant of both mouse and piglet infection [25,42]. To verify the colonization defect observed from a pool of transposon mutants, a targeted knockout mutant was created for *cheO* in *C. jejuni* 81–176 and the mutation of *cheO* did not affect *C. jejuni* growth in rich brain heart infusion (BHI) liquid medium (S2 Fig). Mouse infection experiments were carried out with equivalent amounts of the ΔcheO mutant and the wild-type strain. Drastically lower numbers of the ΔcheO mutant were recovered from the mouse gut in competition with the wild-type strain (Fig 1A). In contrast, complementation of *cheO* in ΔcheO mutant can restore the phenotype comparable to the wild-type strain (Fig 1A). Therefore, CheO is required for *C. jejuni* mouse colonization.

Sequence analyses of CheO did not identify any known domain or motif that provided a clue for its cellular function. In addition, this protein does not contain any obvious signal peptide or transmembrane region, most likely to be a cytoplasmic protein. In the genome of *C.*

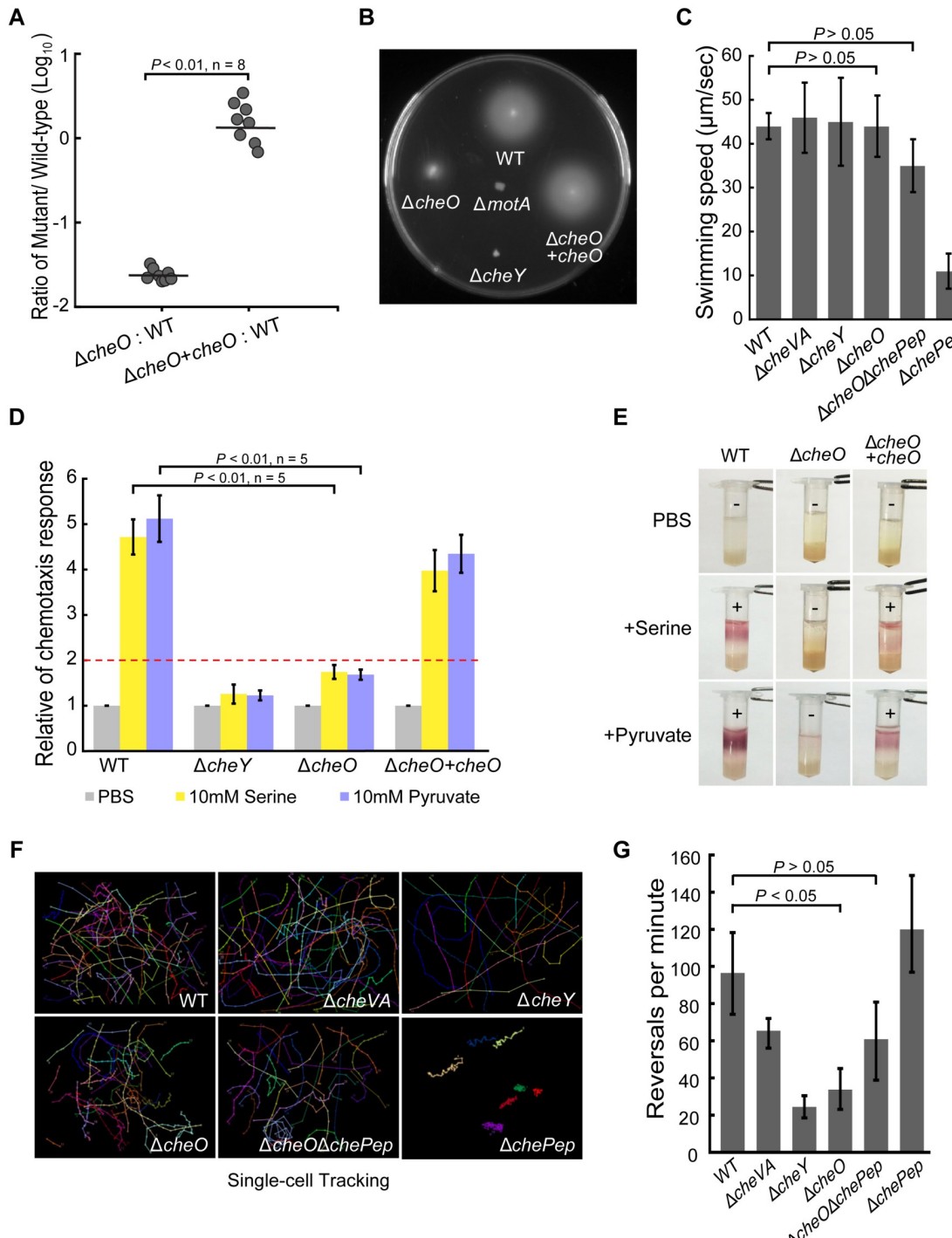

**Fig 1. CheO affects *C. jejuni* chemotaxis. (A)** Role of CheO in mouse colonization. Mice were inoculated with an equal number of wild-type *C. jejuni* and Δ*cheO* mutant or complemented mutant strains via oral gavage (n = 8). Ratio was calculated by the colony-forming units (CFUs) of the mutant over the CFUs of the wild-type strain recovered from the ceca of infected mice. Statistical significance was determined by a one-way ANOVA, P < 0.01. **(B)** Soft agar motility assay of *C. jejuni* wild-type and Δ*cheO*, Δ*motA*, Δ*cheY* mutants. **(C)** Quantification of swimming speed of *C. jejuni* wild-type, Δ*cheVA*, Δ*cheY*, Δ*chePep*, Δ*cheO* and Δ*cheO*Δ*chePep* mutants. Data are shown as mean ± SEM. The differences between Δ*cheO* mutants and wild-type were tested using Student's *t*-test. **(D)** Capillary chemotaxis assay of *C. jejuni* wild-type and mutant strains. 10 mM serine or pyruvate in PBS solution serves as the chemoattractant and PBS buffer as a negative control. The histogram shows the relative chemotaxis response (RCR) of mutants relative to wild-type strain. RCR≥2 indicated a positive chemotaxis response. Data are shown as the mean ± SD (n = 5). *P*-

values derived from Student's *t*-test for Δ*cheO* mutant compared with wild-type strain are shown on the column. **(E)** Tube-based chemotaxis assay of *C. jejuni* wild-type and *cheO* mutant strains. The + and–signs indicate the presence or absence of chemotaxis behavior towards the compounds. **(F)** Single-cell tracking of *C. jejuni* wild-type and mutant strains in BHI medium with the microscope slides and coverslips sealed in microaerobic atmosphere. Three individual experiments were performed for each strain and 20–30 cells were tracked in each experiment. The images shown here are representatives for each strain to compare their swimming behavior. **(G)** Quantification of reversal rates of *C. jejuni* wild-type and mutant strains. The reversal rates were calculated as the number of directional switches per minute per cell and data are shown as mean ± SEM. Differences between the Δ*cheO* mutants and the wild-type were statistically analyzed by Student's *t*-test.

*jejuni* 81–176, *cheO* is a stand-alone gene with 86bp intergenic region to its upstream *guaA* (CJJ81176_1264) and 240bp to its downstream *purD* (CJJ81176_1266). As a phenotype screen for the Δ*cheO* mutant, motility assays were performed first due to the prominence of chemotaxis and motility in *C. jejuni* host colonization. The results showed that mutation of *cheO* had reduced, rather than abolished the ability of *C. jeuni* to swarm on the soft agar plate (Fig 1B), but did not affect its swimming ability in liquid (S1 Movie). The swarming defect of Δ*cheO* mutant on soft agar is not due to growth deficiency (S2 Fig) nor reduced swimming speed (Fig 1C), which suggests the possibility of impaired chemotaxis. To test this, both capillary and tube-based chemotaxis assays were carried out and Δ*cheO* mutant showed significant chemotactic defects toward the *Campylobacter* carbon source, serine and pyruvate, compared to the wild-type strain (Fig 1D and 1E). The reintroduction of a wild-type copy of *cheO* elsewhere in the chromosome of Δ*cheO* mutant restored a phenotype comparable to the wild-type strain. This suggested that the chemotactic defects were caused solely by the mutation of *cheO* (Fig 1B–1E).

To further confirm the effect of CheO on chemotaxis, the swimming behavior of the wild-type strain, Δ*cheO* mutant and other chemotaxis gene mutants were examined by single-cell tracking. The *C. jejuni* wild-type strain switched swimming direction at an average rate of 95 reversals/min, whereas the Δ*cheO* mutant showed a significantly reduced reversal rate (30 reversals/min), similar to the behavior of Δ*cheY* mutant (22 reversals/min) (Fig 1F and 1G). Notably, *C. jejuni* also encodes a ChePep homolog (CJJ81176_1193), which was first reported in *H. pylori* but not yet characterized in other species [32,33]. The *C. jejuni* Δ*chePep* mutant showed a hyperreversal rate of 120 reversals/min compared to the wild-type strain (Fig 1F and 1G), which was similar to the *H. pylori* Δ*chePep* mutant. Importantly, the Δ*cheO*Δ*chePep* double mutant abolished the hyperreversal swimming of the Δ*chePep* mutant, indicating that a loss of CheO is epistatic over the Δ*chePep* (Fig 1F and 1G). Taken together, the Δ*cheO* mutant displayed straight swimming patterns without affecting swimming speed (Fig 1C, 1F and 1G), suggesting a role of CheO in controlling directional switching of flagellar rotation.

## Polar localization of CheO depends on chemosensory array formation

*C. jejuni* possess one flagellum at each pole and electron cryotomography demonstrated that collar-shaped chemoreceptor arrays enclose the two cellular poles of *C. jejuni* and are immediately adjacent to and surrounding the flagellar motor [43–45]. We reasoned that if CheO plays a direct role in chemotaxis, it should localize at the cellular poles as do other chemotaxis proteins. To visualize the subcellular distribution of CheO, the *cheO*-sfGFP fusion gene was constructed and integrated into the *C. jejuni* chromosome to replace native *cheO* to ensure the native level of expression. CheO-sfGFP is functional since the *C. jejuni* strain expressing this fusion gene was able to spread on soft agar in a manner similar to the wild-type strain (S3 Fig). Fluorescence microscopy showed that CheO-sfGFP localizes exclusively to the poles of *C. jejuni* under microaerobic condition (Fig 2A).

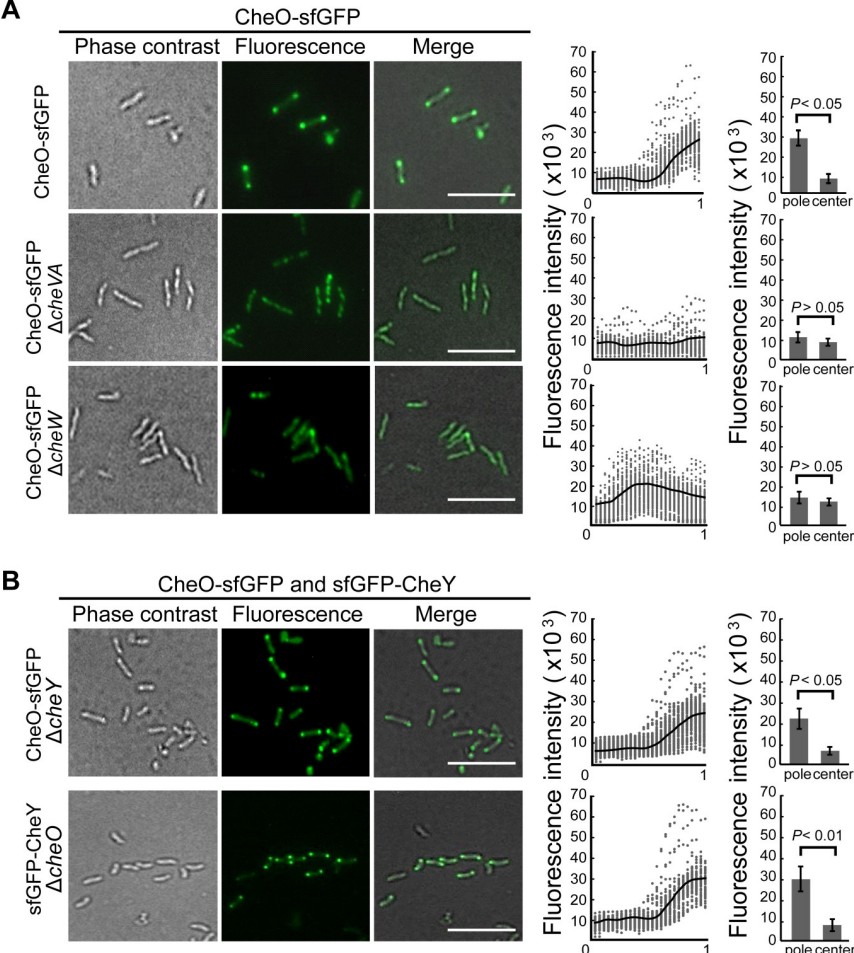

**Fig 2. The localization of CheO in *C. jejuni*. (A)** Fluorescence microscopy of the intracellular localization of CheO-sfGFP in *C. jejuni* wild-type, Δ*cheVA* and Δ*cheW* mutant strains. The scatter plot diagram shows the fluorescence intensity of CheO-sfGFP distributed along with the axe of 100 individual cells from the center (0) to the pole (1.0). The black line represents the average intensity of each measuring point. The histogram shows the quantification of the CheO-sfGFP signal intensity at the pole (1.0) and the center (0) of each cell. Data presented as mean ± SEM. Scale bar, 5 μm. **(B)** The localization of sfGFP-CheY in Δ*cheO* mutant and CheO-sfGFP in Δ*cheY* mutant.

To further examine whether the polar localization of CheO is dependent on other chemotaxis proteins, the *cheO*-sfGFP fusion gene was introduced into *C. jejuni* mutants with deletions of the core chemotaxis genes *cheVA*, *cheW*, or *cheY*. In the absence of CheVA or CheW, CheO-sfGFP lost its polar fluorescence loci, indicating that the proper localization of CheO requires both kinase CheA and scaffolding proteins CheVW (Fig 2A). In contrast, CheO-sfGFP still localizes at the cellular poles in the absence of CheY, which is a diffusing signaling protein between the chemosensory array and flagellar structures (Fig 2B). The localization of CheY was examined through the introduction of sfGFP-*cheY* fusion gene in both the wild-type strain and the Δ*cheO* mutant. sfGFP-CheY displayed a bipolar localization pattern in both strains, therefore CheO and CheY localize at the poles independent of each other. As demonstrated in *E. coli*, chemoreceptor array formation requires CheA and CheW but not CheY [29,46]. These results in *C. jejuni* suggest that the polar localization of CheO is dependent on chemosensory array formation.

## Oxygen level and Tlps involved in energy taxis affect CheO localization

During microscopic observations in a normal atmosphere, we noticed that the polar localization of CheO was less stable than that of CheY. Except for the microscopic observation step, all the above *in vitro* experimental characterizations of *C. jejuni* wild-type strain and various mutants were performed in a hypoxia workstation with an atmosphere of 5% $O_2$ and 10% $CO_2$ at constant concentrations, which is the preferred microaerobic condition for *C. jejuni*. CheO-sfGFP showed bright bipolar fluorescence loci only if the BHI liquid medium used to resuspend *C. jejuni* cells was equilibrated with the air in the hypoxia workstation and the cover slide was fully sealed before removing it from the station to maintain the cells under a microaerobic condition. Otherwise, the fluorescence intensity at the poles significantly decreased and dispersed, indicative of mislocalization or instability of CheO-sfGFP. We suspect that the fluorescence change of CheO-sfGFP in *C. jejuni* cells might be related to oxygen variation. To test this, the location of CheO-sfGFP was examined in *C. jejuni* cells that were cultured under microaerobic condition but washed with aerobically equilibrated BHI medium or directly cultured in a normal atmosphere incubator with a constant 10% $CO_2$ supplement. In these two experimental setups that were referred to aerobic conditions, CheO-sfGFP did not display bipolar fluorescence loci, in contrast to the circumstance that *C. jejuni* cells were grown and handled completely in a microaerobic workstation (Fig 3A). As a control, the bipolar distribution pattern of sfGFP-CheY was not altered under aerobic conditions (Fig 3B). In addition, the Δ*cheO* mutant showed a less noticeable swarming defect in soft agar incubated under aerobic than microaerobic conditions compared to the wild-type strain on the same plate, suggesting that CheO has a greater requirement for chemotactic motility under microaerobic conditions (Fig 3C and 3D). The expression levels of *cheO* in *C. jejuni* grown under both aerobic and microaerobic atmospheres were examined through qRT-PCR and no significant differences were observed (Fig 3E). Therefore, the polar localization of CheO is responsive to environmental oxygen variation, but not likely to be regulated at the transcriptional level.

In bacteria, oxygen can be sensed by several sensory domains including Per-Arnt-Sim (PAS), heme-bound globin/GAF/BLUF, hemerythrin, Oxygen-binding di-iron protein (ODP) domains and proteins with [Fe-S] clusters coordinated by cysteine residues [47–49]. CheO does not have any known sensory domain and sequence alignment of CheO homologs did not yield any conserved cysteine residues (S4 Fig). Based on these data, CheO appears less likely to be able to sense oxygen or other environmental stimuli directly, and other sensory components in the chemosensory array might affect its localization and function. Since we already examined the role of CheVAWY on CheO localization, we then tested the Tlps that constitute the sensory repertoire for chemotaxis. The *cheO*-sfGFP fusion gene was introduced into *C. jejuni* mutants that were deleted with each of 8 *tlp* genes except *tlp3* and *tlp5*, which are pseudogenes in *C. jejuni* 81–176 (S1 Fig). Microscopic observation showed bipolar fluorescence of CheO-sfGFP in all mutants, indicating that none of the Tlps alone can determine the polar localization of CheO (S5 Fig). Tlp9 (CetA) and Tlp6 (homolog of TlpD in *H. pylori*) mediate energy taxis involved in sensing the energy levels and oxidative stress [38–40,50]. Tlp9 does not contain a PAS domain but it interacts with two sensor proteins Aer1 and Aer2, both of which are composed of the PAS domain alone (S1 Fig) [39,40]. Tlp6 is a cytoplasmic chemoreceptor with a CZB domain but no PAS domain (S1 Fig) [50,51]. A quadruple knockout mutant Δ*tlp6*Δ*tlp9*Δ*aer1*Δ*aer2* was generated and in this background, CheO-sfGFP dispersed throughout the cells whereas sfGFP-CheY still displayed a bipolar distribution (Fig 3F). In addition, we tested the motility behavior of the quadruple mutant Δ*tlp6*Δ*tlp9*Δ*aer1*Δ*aer2* to examine whether the mutations of four chemoreceptor genes will destabilize the chemosensory array. The swarming halo of Δ*tlp6*Δ*tlp9*Δ*aer1*Δ*aer2* was slightly reduced on the soft agar plate, but its

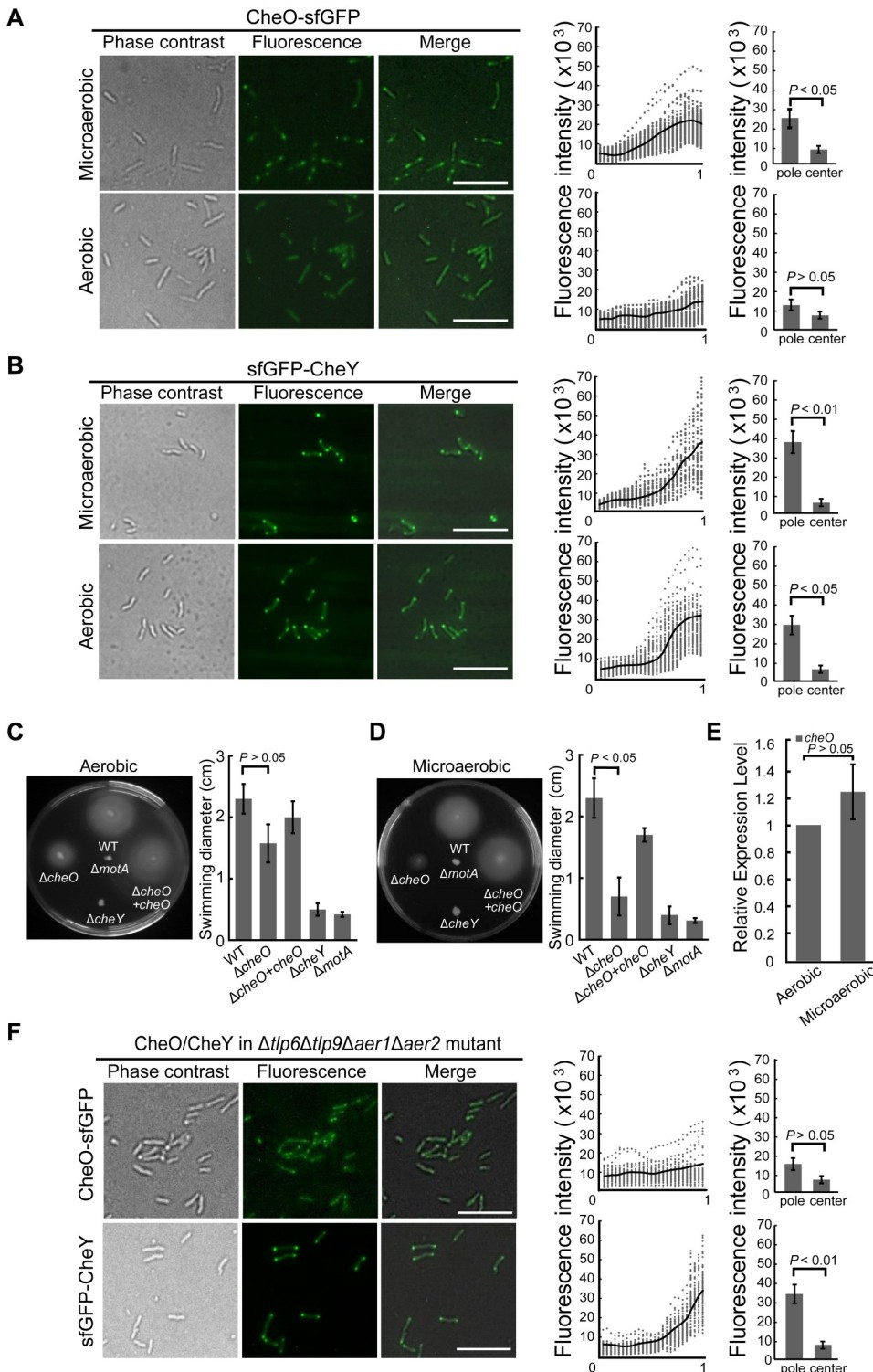

**Fig 3. The oxygen level and Tlps in energy taxis influence CheO localization. (A-B)** Different oxygen level alters CheO-sfGFP localization and sfGFP-CheY as a control. **(C-D)** Soft agar motility assay of *C. jejuni* wild-type and mutant strains in aerobic/microaerobic conditions. Histogram graph shows the diameter of the swimming zone of mutant strains in comparison with the wild-type, with Δ*cheY* and Δ*motA* mutants as negative controls. Data are shown as the mean ± standard deviation (n = 6). *P*-values derived from Student's *t*-test are shown on the column. **(E)** Transcription profile of *cheO* after growth in aerobic or microaerobic conditions for 24 hours. Data represent the average of three experiments and error bars show SD. **(F)** Localization of CheO-sfGFP and sfGFP-CheY in Δ*tlp6*Δ*tlp9*Δ*aer1*Δ*aer2* mutant. The scatter plot graph and histogram are the same as Fig 2.

swimming velocity and reversal frequency were not significantly different from the wild-type strain in the single-cell tracking analysis (S6 Fig). These results suggested that the quadruple mutant retains chemosensory array structure. Therefore, more than one Tlps, particularly those involved in energy taxis, coordinately determine the recruitment of CheO to the chemo-sensory arrays at the cellular poles.

## CheO interacts with multiple chemotaxis and flagellar rotor components

To determine how CheO assists chemotactic motility, bacterial two-hybrid (BTH) assays were conducted to examine whether CheO interacts with any of the chemotaxis signaling proteins CheV/A/W/Y/X/Z/Pep and flagellar rotor components FliG/M/N/Y. The results showed that CheO likely interacts with three chemotaxis proteins CheA, CheY, CheZ, and two flagellar proteins FliM and FliY (Fig 4A). These positive potential interactions were further verified by *in vitro* pull-down assays. Only interactions between CheO and CheA, CheZ, FliM, FliY were confirmed but not with CheY (Fig 4B). The fluorescence microscopy studies showed that CheO and CheY localize at the cellular poles independent of each other (Fig 2B), therefore they may function independently and their interaction in the BTH assay might be a false posi-tive result.

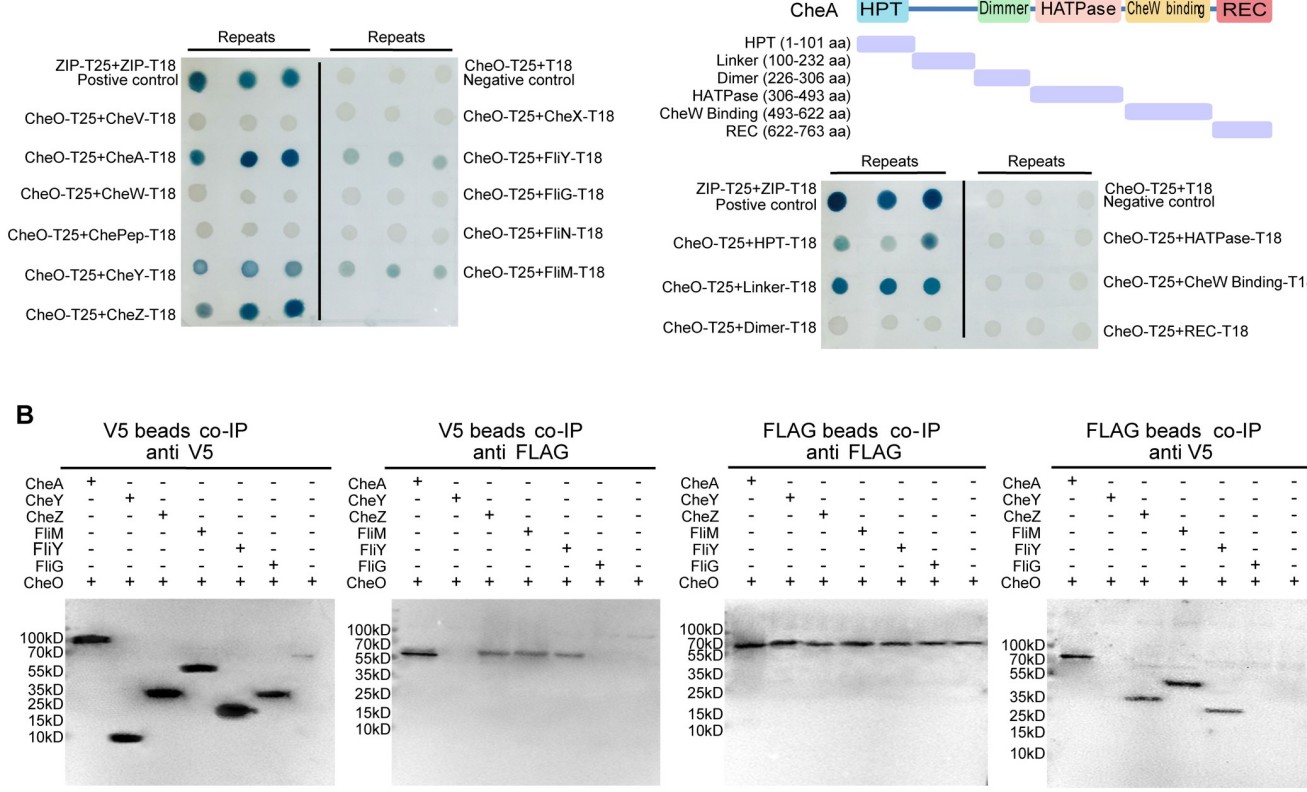

**Fig 4. Interactions between CheO and multiple chemotaxis and flagellar rotor components. (A)** BTH analysis for interactions between CheO and chemotaxis & flagellar components. The formation of blue colonies shows that a protein-protein interaction occurs. **(B)** Pull-down assays of CheO with potential interacting proteins from BTH analysis. CheO was fused with 3X FLAG-tag and the interacting candidates were fused with V5-tag. V5-tag beads co-IP products were verified by western blotting using primary antibodies against V5-tag (bait) and FLAG-tag (prey); FLAG-tag beads co-IP products were also verified by western blotting using primary antibodies against FLAG-tag (bait) and V5-tag (prey). FliG serves as a negative control. **(C)** BTH analysis for CheO and each domain of CheA. The purple fragments indicate the truncated fragments of CheA.

Particularly, the CheA homolog in *C. jejuni* is a hybrid of kinase CheA with an additional receiver domain at the C-terminus. Because protein–protein interaction studies showed that CheO interacts with CheA, we wondered how they interact with each other. CheA is a multi-domain protein and each well-defined domain was cloned into BTH vectors that were then examined for their interactions with CheO. Interestingly, only the N-terminal histidine-containing phosphotransfer (HPT) domain and the linker domain of CheA showed potential interactions with CheO (Fig 4C). The interaction pattern between CheA and CheO is similar to CheA and CheY interaction, in which the linker domain of CheA recruits CheY then the HPT domain transfers the phosphoryl group to CheY [52,53].

Since the polar localization of CheO is lost in quadruple knockout mutant $\Delta tlp6\Delta tlp9\Delta aer1\Delta aer2$, we further examined whether CheO interacts with any of the chemoreceptors through BTH assay. As shown in S7 Fig, CheO does not interact with any of the energy taxis receptor Tlp6, Tlp9, Aer1, Aer2, and the other Tlps (Tlp1/4/7/8/10). Besides, since the above results suggested CheO directly interacts with two flagellar rotor components FliM and FliY, we investigated whether these flagellar proteins affect CheO localization. A copy of *cheO-sfGFP* was introduced into $\Delta fliMY$ mutant to replace the native *cheO*, and fluorescence imaging showed bipolar distribution of CheO-sfGFP in *C. jejuni* cells in the absence of FliM and FliY (S8 Fig).

## Suppressor analysis of *C. jejuni* $\Delta cheO$

To further explore the role of CheO in chemotactic control of flagellar rotation, spontaneous suppressor mutations were isolated that significantly increased the spreading phenotype of the $\Delta cheO$ mutant on soft agar. $\Delta cheO$ mutants were inoculated microaerobically on soft agar plates at 37°C for up to four days. Subsequently, four independent suppressor mutants were collected from the periphery of the motile halo originating from different inoculation points. These four mutants showed increased spreading compared to the parental $\Delta cheO$ mutants on soft agar (Fig 5A). Genomic sequencing revealed three classes of suppressor mutations (Table 1).

A single nucleotide alteration producing FliM[L99F] is found in all four suppressor mutants. The L[99] residue is conserved and located in the middle domain of FliM, specifically in the loop between the α2-helix and β2-strand of FliM, which might be a site for protein–protein interactions [54] (Fig 5B). In addition, a single nucleotide variation in *cheA* producing CheA[P752L] is found in one of the four suppressor mutants. P[752] is within a conserved KPF motif in the REC domain of CheA that is important for dimerization in the classical REC domain. A frameshift deletion within the *cstA* gene that encodes a transporter for peptide uptake is also present in all four suppressor mutants. We replaced the wild-type *fliM* gene with *fliM*[G297T] (nucleotide mutation) in the chromosome of $\Delta cheO$ mutant to test whether FliM[L99F] (the corresponding amino acid mutation) can alleviate the spreading defect in the absence of CheO. This mutant demonstrated a spreading phenotype comparable to the wild-type level, indicating that the L99F alteration in FliM alone was sufficient to rescue the flagellar rotation to wild-type level in the absence of CheO (Fig 5C). In addition, we tested whether FliM[L99F] is specific to $\Delta cheO$ and the results showed that FliM[L99F] cannot increase the swarming motility of the wild-type strain or restore the phenotype of $\Delta cheVA$ mutant (S9 Fig). The mutations in *cheA* and *cstA* genes were not examined further.

## CheO is functionally conserved in host-associated and few free-living *Campylobacterota* species

Bioinformatic analyses for potential CheO homologs revealed that the protein is only present in species within the phylum *Campylobacterota*. The species containing the CheO homolog

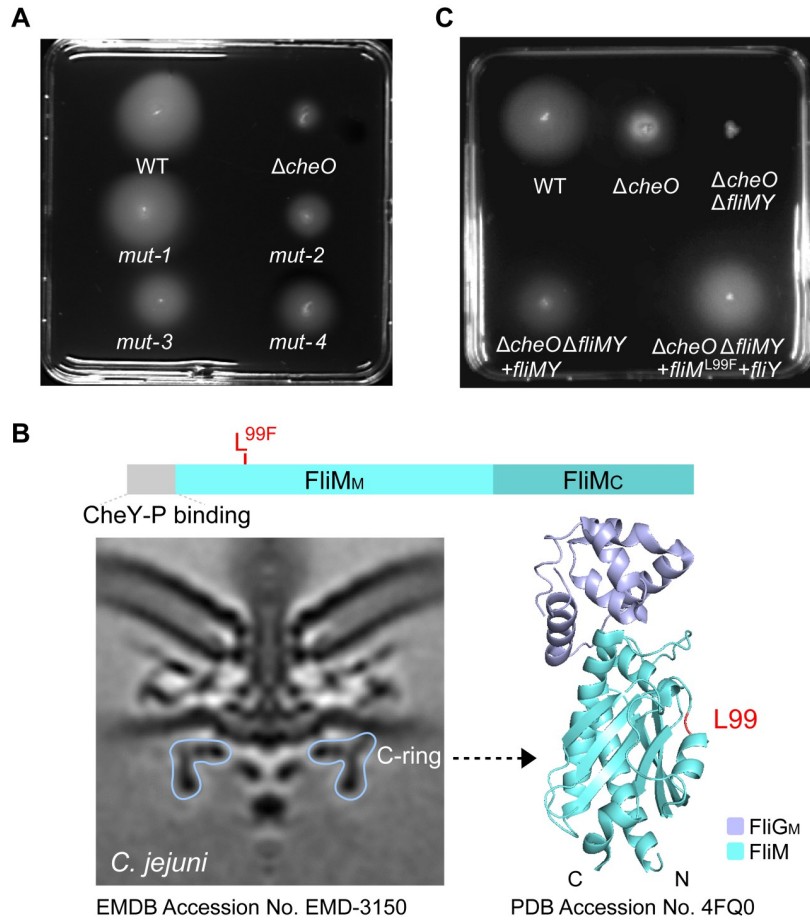

**Fig 5. Analysis of *C. jejuni* Δ*cheO* suppressor mutants in microaerobic condition.** (**A**) Soft agar motility assay of *C. jejuni* wild-type, parental Δ*cheO* mutant and Δ*cheO* isogenic suppressor mutants "mut-1" to "mut-4". (**B**) The C-ring of *C. jejuni* flagellar motor, domain architecture and X-ray structure of FliM. FliM$_M$: the middle domain of FliM; FliM$_C$: the C terminal domain of FliM [45,54]. (**C**) Soft agar motility phenotypes of *C. jejuni* wild-type and the relative isogenic mutants in microaerobic conditions.

**Table 1. Identification of Δ*cheO* suppressor mutations in *C. jejuni* 81–176.**

|  | Sample[a] | Gene | Protein | Type | Mutation [b] | Variant Frequency (>40%) |
|---|---|---|---|---|---|---|
| **Class I** | mut-1 | *CJJ81176_0098* | FliM | SNV | *fliM*$^{G297T}$ (protein L$^{99F}$) | 0.49 |
|  | mut-2 | *CJJ81176_0098* | FliM | SNV | *fliM*$^{G297T}$ (protein L$^{99F}$) | 0.95 |
|  | mut-3 | *CJJ81176_0098* | FliM | SNV | *fliM*$^{G297T}$ (protein L$^{99F}$) | 0.99 |
|  | mut-4 | *CJJ81176_0098* | FliM | SNV | *fliM*$^{G297T}$ (protein L$^{99F}$) | 1 |
| **Class II** | mut-1 | *CJJ81176_0310* | CheA | SNV | *cheA*$^{C2255T}$ (protein P$^{752L}$) | 0.47 |
| **Class III** | mut-1 | *CJJ81176_0924* | CstA | INDEL | frameshift deletion at bases of 669–682 (GCACAAAACCTATT) | 0.86 |
|  | mut-2 | *CJJ81176_0924* | CstA | INDEL | frameshift deletion at bases of 669–682 (GCACAAAACCTATT) | 0.99 |
|  | mut-3 | *CJJ81176_0924* | CstA | INDEL | frameshift deletion at bases of 669–682 (GCACAAAACCTATT) | 1 |
|  | mut-4 | *CJJ81176_0924* | CstA | INDEL | frameshift deletion at bases of 669–682 (GCACAAAACCTATT) | 1 |

[a] Suppressor mutants mut-1 to mut-4 were isolated from *C. jejuni* 81–176 Δ*cheO*.

[b] The mutation sites were identified by genomic sequencing.

are mainly from host-associated genera *Campylobacter*, *Helicobacter*, *Wolinella*, and two free-living genera *Hydrogenimonas* and *Sulfuricurvum*, but not from any other bacterial phyla (S10 Fig). Notably, not all species from *Campylobacter* and *Helicobacter* possess the CheO homolog, but those that have the homolog tend to have at least one of the Tlps (Tlp6, 8, 9) mediating energy taxis (S10 Fig). To assess whether these identified homologs are functionally conserved, homologous genes from the zoonotic pathogen *Helicobacter pullorum* and distantly related extremophiles *Hydrogenimonas thermophila* were introduced into the *C. jejuni* Δ*cheO* mutant, respectively. The expression levels of CheO homologs were verified by immunoblotting for a FLAG-tag fused to each foreign CheO at the C-terminus (Fig 6A). Despite the differences in protein length and sequence, CheO homologs of *H. pullorum* and *H. thermophila* recovered the chemotactic motility of the *C. jejuni* Δ*cheO* mutant (Fig 6B and 6C). These results suggest that CheO is functionally conserved in diverse species of the *Campylobacterota* phylum.

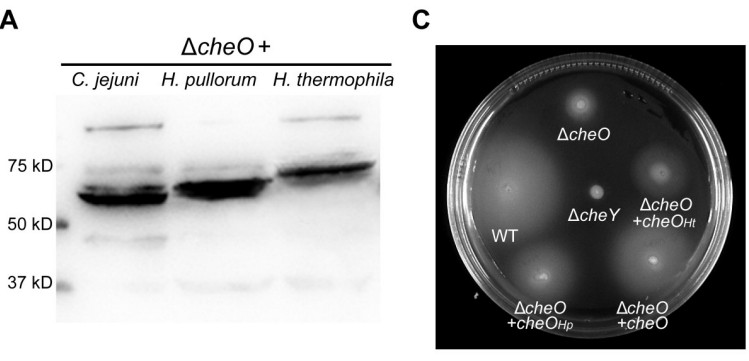

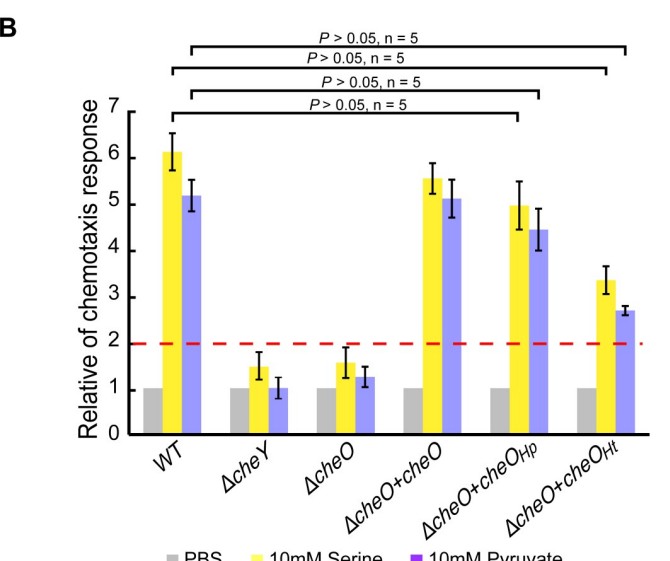

**Fig 6. Functional conservation of CheO in *Campylobacterota* species. (A)** Immunoblotting of CheO homologs from *C. jejuni*, *Helicobacter pullorum* and *Hydrogenimonas thermophila* that were fused with 3X FLAG-tag and expressed in *C. jejuni* Δ*cheO* mutant. The blot was probed with anti-FLAG antibodies. **(B)** Capillary chemotaxis assays of *C. jejuni* wild-type and Δ*cheO* mutant strains complemented with various CheO homologs. Data are shown as the mean ± SD (n = 5). The differences between the *cheO* isogenic mutants and wild-type were tested using Student's *t*-tests. **(C)** Soft agar motility assay of *C. jejuni* wild-type and Δ*cheO* mutant strains complemented with various CheO homologs. Δ*cheY* mutant as a negative control.

## Discussion

Chemotactic motility is crucial for *C. jejuni* host colonization and its chemotaxis system is more complicated than the *E. coli* paradigm. A classification scheme of the chemosensory system based on phylogenomic markers has assigned the "F3 class" to the chemotaxis pathway in *C. jejuni*, *H. pylori*, and other *Campylobacterota* species; whereas the *E. coli* chemotaxis belongs to the "F7 class" [55]. Our recent evolutionary analyses of the chemosensory system in the phylum *Campylobacterota* revealed that the F3 class evolved in the common ancestor of this phylum and was passed on to all descendants including the host-associated *C. jejuni* and *H. pylori* [56]. In addition, a distinctive feature of the F3 class is that their transmitter genes are highly dispersed in the genomes with nine genes *cheVAW/BR/X/Y/Z/Pep* in six loci, in contrast to other chemosensory classes that generally have all of their transmitters encoded in one gene cluster [57]. Because of the scattered genomic distribution, it is difficult to predict and identify new components for the F3 class that do not have homology to known chemotaxis proteins. In this study, we identified a novel chemotaxis protein CheO in *C. jejuni*, whose homologs are functionally conserved in, and restricted to, the *Campylobacterota* phylum, adding an element to the complex F3 class. The discovery of CheO here in conjunction with the *Campylobacterota*-specific ChePep suggests that there might be other chemotaxis components functioning in the F3 class, whose complexity in signal transduction is not yet fully understood.

Chemotaxis assays confirmed that CheO affects *C. jejuni* chemotactic behavior. This is likely the reason that it is required for host colonization. CheO localizes at the cell poles where the chemosensory array and flagellum are positioned in *C. jejuni*. In addition, the polar localization of CheO depends on chemosensory array formation since the absence of CheVA results in dispersion of CheO away from the cell poles. Protein–protein interaction assays indicated that CheO can interact with chemotaxis proteins CheA/CheZ and flagellar rotor proteins FliM/FliY, suggesting a direct role of CheO in chemotactic control of rotor switch. Suppressor screening of the Δ*cheO* mutant also identified a mutation in FliM that can rescue the chemotactic defect caused by the absence of CheO, supporting the functional linkage of CheO and flagellar rotation. Notably, although both FliM and FliY interact with CheO but they do not affect the polar location of CheO, implying that flagellar rotor might be a regulating target for CheO. Based on these results, we suspect that CheO can transfer chemotaxis signals to the flagellar rotor, independent of CheY and epistatic to ChePep. Furthermore, CheO is likely to associate with CheA within the chemosensory array in a manner similar to CheY but our *in vitro* phosphorylation assays did not yield conclusive results. Thus, the detailed mechanism of CheO in chemotaxis signal transduction is unclear.

Environmental oxygen level influences the localization and function of CheO in *C. jejuni*. Only if *C. jejuni* cells were cultured and handled strictly in a microaerobic chamber, CheO displayed polar localization and the Δ*cheO* mutant showed more severe swarming defect on soft agar. Under aerobic conditions, CheO was dispersed throughout the cell. Sequence analyses of CheO did not identify any sensory domain or conserved cysteine residue, suggesting that this protein is less likely to sense oxygen directly. The chemosensory repertoire was further examined and none of the Tlps alone could determine the CheO localization. Besides, CheO does not interact with any chemoreceptors directly in our BTH assays. Instead, Tlp6 and Tlp9/Aer1/Aer2 that mediate energy taxis together affect the polar location of CheO. Changes in oxygen availability can regulate the energy status of *C. jejuni* by the re-modeling of its electron transport chain and metabolic substrates [12,40]. Therefore, oxygen concentration may not be the only factor or the direct factor that affects CheO localization. Tlp6 with CZB domain and Aer1/Aer2 with PAS domain may monitor the internal metabolic and redox status upon oxygen fluctuation [38–40,50]. Then, these receptors and other Tlps can transduce the signals to

affect the chemosensory array conformation and determine the recruitment of CheO to the chemosensory array. Moreover, because CheO is more necessary in microaerobic than in aerobic conditions, this protein likely promotes more vigorous chemotactic motility to help *C. jejuni* sense and relocate to niches with preferred oxygen level and other favorable conditions.

Energy taxis confers a fitness advantage to *Salmonella enterica* serovar Typhimurium during inflammation in a mouse colitis model [58,59]. Since *C. jejuni* causes more cases of human colitis than *S.* Typhimurium in developed countries, it is worthwhile to examine whether energy taxis and the CheO identified here play a role in *C. jejuni* adaptation in the inflamed host gut using recently developed diarrhea model for *C. jejuni*. Taken together, the identification of a novel chemotaxis protein CheO provides new insights into the complexity of chemotaxis behavior and host adaptation of *C. jejuni*. More mechanistic studies are needed to pinpoint the unique role of CheO in *C. jejuni* and closely related pathogens.

## Materials and methods

### Ethics statement

All animal experiments were conducted according to protocols approved by the Southern Medical University Institutional Animal Care and Use Committee (China). The mice (male and female) were treated with multiple antibiotics via drinking water for 4 weeks to eliminate their commensal gut flora. After the infection of *C. jejuni* wild-type and mutant strains, all mice were sacrificed by euthanasia using the $CO_2$ gradual fill method, and the cecum was harvested immediately.

### Bacterial strains and culture conditions

The list of strains, plasmids, primers and related antibiotics are listed in S1 and S2 Tables. The *C. jejuni* 81–176 wild-type and mutant strains were routinely grown on blood agar plates (Trypticase soy agar supplemented with 5% sheep blood) at 37˚C in BACTROX-2 microaerobic workstation (SHELLAB, USA) equilibrated to a 5% $O_2$ and 10% $CO_2$ atmosphere. For liquid cultures, *C. jejuni* strains were grown in BHI medium. Soft agar motility assays were also performed in an incubator equilibrated to a 10% $CO_2$ atmosphere where the oxygen level is only slightly lower but comparable to normal atmosphere, referred to aerobic condition in this study. The *C. jejuni* mutants were selected on brucella broth agar plates supplemented with antibiotics as indicated below. *E. coli* was grown on LB medium or agar plates at 37˚C under aerobic conditions. The selection medium contained antibiotics at the following concentrations: chloramphenicol-50 μg/mL for *E. coli* and 10 μg/mL for *C. jejuni*; kanamycin-50 μg/mL; apramycin-50 μg/mL; ampicillin-100 μg/mL. All *C. jejuni* strains were stored at -80˚C in BHI medium with 30% glycerol, and *E. coli* strains were stored at -80˚C in LB medium with 15% glycerol.

### *C. jejuni* mutant construction and complementation

*C. jejuni* 81–176 knockout mutant strains were constructed by the gene insertion or replacement strategy, in which an antibiotic resistance cassette was inserted into the open reading frame (ORF) of the target gene as previously described [30]. The upstream and downstream regions of the target gene (approximately 1kb each fragment) were PCR amplified and introduced a BamHI or EcoRI restriction enzyme cutting site in the middle. The two fragments were fused into the linearized pBluescript II SK plasmid following the Gibson assembly protocol [60]. The resulting plasmid was digested with BamHI or EcoRI enzyme, then a kanamycin or apramycin gene cassette was inserted by T4 ligase. The recombination plasmids were

transform to *E. coli* DH5α and transformants were selected on LB plates containing kanamycin or apramycin. All plasmids were verified by DNA sequencing and were naturally transformed into *C. jejuni* 81–176 for gene allelic exchange. The transformants were selected on Brucella broth agar plates with kanamycin or apramycin. The mutation was confirmed by PCR analysis and DNA sequencing.

The *C. jejuni* gene knockout mutants were complemented by inserting the wild-type copy of the target gene into the *hsdR* locus with a chloramphenicol resistance cassette and a 3XFLAG tag fused to the target gene as previously described [61]. The complemented mutants were selected on Brucella broth agar plates with chloramphenicol and kanamycin/apramycin. PCR tests were used to verify the recombinant gene regions of all constructs. The sequences of CheO homologs from *Helicobacter pullorum* (accession no.: WP_104745726.1) and *Hydrogenimonas thermophila* (accession no.: WP_092912878.1) were downloaded from NCBI GenBank and synthesized by Sangon Biotech (China).

## Mouse colonization competition assay

All animal experiments were conducted according to protocols approved by the Southern Medical University Institutional Animal Care and Use Committee (China). Six- to eight-week-old C57BL/6 mice (male and female) were treated with multiple antibiotics via drinking water for 4 weeks to eliminate their commensal gut flora for *C. jejuni* infection as previously described [25]. All antibiotics were removed from the drinking waters 2 days before the infection. Mid-log phase *C. jejuni* wild-type and mutant strains were collected by centrifugation (10 min, $3,000 \times g$), washed once with sterile phosphate-buffered saline (PBS), and resuspended at $10^9$ CFUs in 100 μL of PBS. The oral administration of 100 μL of sodium bicarbonate through stomach gavage was performed first to neutralize the stomach pH, then followed by the oral gavage of $10^9$ CFUs of *C. jejuni* strain competition mixture in 100 μL of PBS. The *C. jejuni* strain competition mixtures contained either wild-type: Δ*cheO* mutant or wild-type: Δ*cheO* +*cheO* complementation mutant in a 1:1 ratio. After 3 days of infection, all mice were sacrificed by euthanasia with $CO_2$. The cecum of each mice was collected and homogenized in 3 mL PBS buffer with 0.05% sodium deoxycholate.

Serial dilutions of bacteria recovered from the cecum homogenates were plated on blood agar plates (3 plates per dilution sample) containing *Campylobacter* selective supplements (Karmali, Oxoid SR0167) and 50 μg/mL kanamycin (for Δ*cheO*) or 10 μg/mL chloramphenicol (for Δ*cheO*+*cheO*). Plates were incubated at 37°C for 2 days before the colonies were counted. The titer of the Δ*cheO* or complementation mutant was obtained from the CFUs recovered on karmali agar plates with kanamycin or chloramphenicol. The titer of the wild-type was calculated by subtracting the number of mutants from the total number of bacteria recovered on karmali agar plates without antibiotics. Finally, the *in vivo* competitive index was calculated for each mouse and corresponded to the ratio of the mutant to the wild-type strain.

## Soft agar motility assay

The *C. jejuni* strains were incubated on blood agar plates for 24 hours in microaerobic conditions at 37°C. A sterilized tip was used to dip the colony, then stabbed on semisolid Brucella broth plates with 0.3% agar. Plates were incubated microaerobically (85% $N_2$, 10% $CO_2$, 5% $O_2$) or aerobically (normal atmosphere supplemented with 10% $CO_2$) at 37°C for 16 hours.

## Capillary and tube-based chemotaxis assays

The *C. jejuni* cultures were inoculated microaerobically on blood agar plates at 37°C overnight. Cells were collected and washed three times with PBS buffer and adjusted to an $OD_{600} = 0.5$.

100 μL suspensions of each strain were added to a 96-well plate. Capillary tubes (1 μL, Drummond Microcap, USA) were filled with PBS buffer containing 20 mM serine and PBS buffer alone as control. All capillary tubes were inserted into bacterial suspension in the 96-well plate. The 96-well plate with capillary tubes was incubated at 37°C in a microaerobic chamber for 15 min. The liquid in capillary tubes was collected and serially diluted in PBS buffer and plated on blood agar for counting CFUs. The relative chemotaxis response (RCR) represented the bacterial counts compared to the control [62]. An RCR greater than 2 was considered a positive chemotaxis response, and differences between the mutants and the wild-type were statistically analyzed by one-tailed Student's $t$-test. For each strain, three replicates were included, and three independent experiments were conducted, ensuring a proper sampling to detect differences between samples.

For tube-based chemotaxis assay, 1 mL $OD_{600}$ = 1.0 of each strain were collected and mixed with 200 μL agar made of PBS buffer and 0.4% agar. The mixture was transferred to the bottom of a 2 mL Eppendorf tube and allowed to solidify for 10 min at room temperature. Samples were overlaid first with 1 mL of PBS agar that was allowed to solidify for 15 min. A sterile piece of filter paper soaked with 10 μL of a 1 M serine in PBS or 10 μL PBS alone was placed on the top and samples were incubated under microaerobic conditions for 24 hours at 37°C. Active bacterial cells that migrated through the upper layer of PBS-agar towards the compound added to the filter paper were visualized by adding 100 μL PBS with 0.01% 2, 3, 5-triphenyltetrazolium chloride (TTC) [63]. The respiratory dye TTC detects redox activity of living cells and results in a red layer in the tube that is visible after 1 hour of incubation.

## Measurement of the swimming behavior

The *C. jejuni* strains were inoculated microaerobically on blood agar plates at 37°C overnight. Cultures were diluted to $OD_{600}$ = 0.2 in 1 mL BHI medium and 1 μL drop was added on a microscope slide, overlaid with a coverslip (CITOTEST Company) for microscopy. Images were captured with a Zeiss Axio Observer A1 microscope equipped with a 100× Plan lens and a Zeiss AxioCam 503 color CCD camera. Images were recorded every 40ms using the acquisition function in the ZEN pro software (Zeiss). Three individual experiments were performed and 20–30 cells were tracked in each experiment. Single-cell tracking was analyzed by the MTrackJ plug-in for ImageJ imaging software (https://imagej.nih.gov/ij/index.html), and the reversal rate was calculated as the number of reversals per minute per cell. Bacterial tracks as XY-coordinates were input into Excel to quantify swimming reversals by calculating vector changes along the swimming trajectory. A reversal was counted as a direction change of >110° with positions between frames [32]. The average reversal rate of each strain was calculated and differences between the mutant and the wild-type were statistically analyzed by a two-tailed Student's $t$-test.

## Fluorescence microscopy

The GPF fusion proteins CheO-sfGFP and sfGFP-CheY were expressed *in situ* in the chromosome of *C. jejuni* wild-type and mutant strains for fluorescence microscopy. Cultures were collected and washed gently in BHI medium equilibrated with the air in the microaerobic workstation and adjusted to an $OD_{600}$ = 0.2. 1 μL of *C. jejuni* culture was sealed in the slides using nail polish in the microaerobic workstation to maintain the low oxygen level. For experiments under aerobic conditions, the BHI liquid medium for washing and resuspending *C. jejuni* cells was prepared with shaking for 24 hours at 37°C in normal atmosphere. Fluorescence microscopy was carried out and a bright-field image was recorded as a control. Images were recorded with a Zeiss Axio Observer A1 microscope equipped with a 100× Plan lens and

a Zeiss AxioCam 503 color CCD camera. At least 5 separate fields of view were counted for each strain. The experiments were repeated for three separate cultures and 100 cells for each strain were counted.

A line scan analysis of the distribution of fluorescence signal intensities along the medial axes of the bacterial cell bodies was carried out using the MicrobeJ plug-in of ImageJ (https://imagej.nih.gov/ij/index.html) software [64]. Random fluorescence and bright field microscopy images were obtained, and cells were defined as regions of interest (ROI) by bright field images. The center within each ROI was identified and 26 measuring points along the medial axes of the bacteria from the center (0) to each of the poles (1.0). Data were presented as the mean of the total number of experiments ± SEM.

## RNA extraction and quantitative real-time PCR

*C. jejuni* 81–176 wild-type were grown under microaerobic and aerobic conditions, separately, and harvested during the mid-exponential phase. RNA extraction was performed using Trizol (TaKaRa, Japan) following the manufacturer's instructions. The purity and concentration of the RNA were determined by gel electrophoresis and a NanoDrop spectrophotometer (Thermo Scientific, USA). RNA was transcribed using the cDNA master Kit (TOYOBO, Japan). Transcript levels were determined with SYBR Green Supermix (TOYOBO, Japan) in a CFX96 Connect Real-Time PCR Detection System (Bio-Rad). The cycling parameters: 95°C for 30 seconds and 40 cycles of 94°C for 15 seconds and 60°C for 30 seconds. The abundance of the *pheX* gene was used as an internal standard and the relative expression levels of *cheO* were calculated using the Quantitation-Comparative CT($2 -\Delta\Delta CT$) method [65].

## BTH analysis

The assays were performed as described to investigate protein-protein interactions [66]. Proteins of interest were fused to T25 or T18 fragments in plasmids pKNT25/ pKT25 or pCH363/ pUT18C. The pair of plasmids expressing fusion proteins with T18 or T25 fragments were co-transformed into *E. coli* strain BTH101 and plated on LB agar containing ampicillin (100 µg/mL) and kanamycin (50 µg/mL). Multiple transformants were inoculated into individual tubes with 0.5 mL of LB broth containing the same antibiotics and 250 mM IPTG. The cultures were incubated for 8 hours at 30°C with shaking. 1 µL of each culture was spotted on LB plates containing the antibiotics, 40 µg/ mL X-Gal and 250 mM IPTG. The plates were incubated at 30°C for 24–48 hours.

## Pull-down and co-IP assay

This experiment utilized the pG0407 plasmid, which has pBAD plasmid as a backbone, with a 3XFLAG tag and a V5 tag with a linker sequence between the two tags. The ORF of *cheO* was amplified and fused with the 3XFLAG tag, and the other tested genes were fused with the V5 tag. These fragments were inserted to the pG0407 vector by Gibson assembly, then transformed into *E. coli* strain BL21 (S1 and S2 Tables). For protein expression, 5 mL overnight culture of each *E. coli* strain was transferred to 100 mL of LB medium with ampicillin 100 µg/mL, and grown to $OD_{600} = 0.6$–0.8. After adding 10 mM arabinose, the cultures were continuously incubated in a shaker at 18°C for 16–18 hours. Bacterial cells were harvested by spinning at 5000 x g, then resuspended in 10 mL of lysis buffer (50 mM Tris, 150 mM NaCl, 0.5%Triton X- 100, 1% PMSF, pH 7.5), and lysed by JN-Mini ultra-high 541 pressure cell disrupters (JNBIO, China). The soluble fractions were collected by centrifugation at 12,000 x g for 10 minutes at 4°C.

V5-tagged and 3XFLAG-tagged proteins were purified with beads following the manufacturer's instructions (Anti-V5-tag mAb-Magnetic Beads, MBL, Catalog Number M167-11; ANTI-FLAG M2 Affinity Gel, Sigma Catalog Number A2220). For the 3XFLAG-tagged CheO protein purification, 20 μL ANTI-FLAG M2 Affinity Gel was added to 3 mL supernatant of this cell lysate and incubated overnight at 4˚C. Collected the beads by centrifugation at 5,000 x g for 10 minutes at 4˚C, and washed the beads three times by 3 mL TBS buffer (50 mM Tris, 150 mM NaCl, pH 7.5). Added 20 μL 3XFLAG peptide eluent (3XFLAG Peptide, Sigma, Catalog Number F4799), mix gently, and incubated for 2 hours at 4˚C and centrifuged at 12,000 x g for 1 minute and collected the elution. For the V5-tagged protein purification, 20 μL Anti-V5-tag mAb-Magnetic Beads was added to 3 mL supernatant of this cell lysate and incubated overnight at 4˚C. Collected the beads by magnetic stand and washed three times with 3 mL TBS buffer (50 mM Tris, 150 mM NaCl, pH 7.5). Resuspend the beads with 30 μL SDS loading dye, boiled for 5 minutes, then centrifuged at 12,000 x g for 1 minute and collected the supernatant. The final elution was loading in 10% SDS-PAGE gel for Western blotting assay by standard procedures. The antibodies used for blotting are the following: anti-V5-Tag Antibody, Proteintech, Catalog Number v5ab; ANTI-FLAG M2 antibody, Sigma, Catalog Number F3165; secondary antibody m-IgGk BP-HRP, Santa Cruz, Catalog Number sc-516102. The blot was then scanned with a CCD scanner (Tanon, China).

## Suppressor analysis of *C. jejuni* Δ*cheO*

To isolate and identify suppressor mutants that restored the motility phenotype of *C. jejuni* Δ*cheO*, Δ*cheO* mutants were streaked on the blood agar plates for 24 hours and then stabbed into Brucella broth semisolid motility agar plates. Bacteria were then incubated for up to 4 days at 37˚C under microaerobic conditions. Potential suppressor mutants were collected from the edge of the biggest motile flares that originated from the point of inoculation of different motility stabs. Then these candidate mutants were examined again on the motility agar plates. Suppressor mutants from each motile isolates were saved. Genomic DNAs from *C. jejuni* wild-type strain, a parental Δ*cheO* mutant, and corresponding suppressor mutant strains were prepared as previously described [67]. Briefly, isolates were grown on blood agar plates at 37˚C under microaerobic conditions. Bacteria were harvested and genomic DNA was isolated using a Qiagen DNeasy kit. Prior to submission for sequencing, the DNA samples were run on a 1.0% agarose gel to check DNA integrity.

Next-generation sequencing library preparations were constructed following the manufacturer's protocol (NEBNext Ultra DNA Library Prep Kit for Illumina). Sequencing was carried out using a 2x150 paired-end (PE) configuration; image analysis and base calling were conducted by the HiSeq Control Software (HCS) + OLB + GAPipeline-1.6 (Illumina) on the HiSeq instrument. Reads obtained for the genomes from the wild type, Δ*cheO* parent strain and suppressor mutants were mapped to the *C. jejuni* 81–176 reference genome (GCA_000015525.1). The Unified Genotyper calls SNV/InDel with GATK (V3.8.1) software. Annotation for SNV/InDel was performed by Annovar. Pindel and CNVnator were used to do genomic structure variation analysis. Single nucleotide polymorphisms (SNVs) with a minimum variant frequency of 40% were identified in regions of the parent. Individual SNVs identified in suppressor mutant genomes were compared to those identified within the wild type and the Δ*cheO* parent genome. SNVs that were unique to the suppressor mutant genomes were supposed to be associated with the suppressor mutant phenotypes.

To reconstruct the FliM$^{L99F}$ suppressor allele on the chromosome of Δ*cheO* mutant, we deleted *fliMY* and then complemented *fliM*$^{L99F}$*Y* as described above in mutant construction. All relevant strains were checked on the soft agar motility agar plates.

### Bioinformatic analysis

The *tlp* genes in the *Campylobacterota* phylum were searched against NCBI RefSeq database [68]. Protein domains were identified using Pfam models, SMART and HHpred searches [69–71]. CheO protein sequence from *Campylobacter jejuni* 81–176 was used as a query in BLAST search against the NCBI RefSeq database with default parameters. The resulting sequences were aligned and edited in MEGA X [72]. The 16S rRNA nucleotide sequences of selected species in *Campylobacterota* were aligned by Clustal W and phylogenetic tree based on this alignment was constructed using the neighbor-joining method implemented in MEGA X, with 1000 bootstraps [73].

## Supporting information

**S1 Fig. Chemoreceptors (also called Tlps) in *C. jejuni* model strains NCTC11168 and 81–176.** Tlp3 in *C. jejuni* 81–176 is annotated as two genes (CJJ81176_1548 and CJJ81176_1549), the same for Tlp7 in NCTC11168 (Cj0591c/Cj0592c).
(TIF)

**S2 Fig. Growth curves of *C. jejuni* wild-type and Δ*cheO* mutants.** All strains were grown at 37˚C in BHI medium in microaerobic conditions.
(TIF)

**S3 Fig. Soft agar motility assay of *C. jejuni* wild-type, *C. jejuni* strains expressing *cheO*-sfGFP or sfGFP-*cheY* at the native *cheO* or *cheY* loci without other mutations, with Δ*motA* mutant as a negative control.**
(TIF)

**S4 Fig. Protein sequence alignment of CheO homologs in species of the *Campylobacterota* phylum.**
(TIF)

**S5 Fig. Fluorescence signal intensity of CheO-sfGFP at cell pole and cell center of *C. jejuni* *tlp* knockout mutants as indicated above the columns.**
(TIF)

**S6 Fig. Motility analysis of *C. jejuni* Δ*tlp6*Δ*tlp9*Δ*aer1*Δ*aer2* quadruple mutant.** **(A)** Soft agar motility assay of *C. jejuni* wild-type, and the Δ*tlp6*Δ*tlp9*Δ*aer1*Δ*aer2* quadruple receptors mutant strain with Δ*motA* mutant as a negative control. **(B)** Single-cell tracking of *C. jejuni* wild-type and Δ*tlp6*Δ*tlp9*Δ*aer1*Δ*aer2* mutant. **(C)** Quantification of swimming speed of *C. jejuni* wild-type and Δ*tlp6*Δ*tlp9*Δ*aer1*Δ*aer2* mutant. Data are shown as mean ± SEM. **(D)** Quantification of reversal rates of *C. jejuni* wild-type and Δ*tlp6*Δ*tlp9*Δ*aer1*Δ*aer2* mutant. Data are shown as mean ± SEM.
(TIF)

**S7 Fig. BTH analysis for interactions between CheO and chemosensory receptors.** The formation of blue colonies shows that a protein-protein interaction occurs and white colonies show negative results.
(TIF)

**S8 Fig. Fluorescence microscopy of the intracellular localization of CheO-sfGFP in *C. jejuni* Δ*fliMY* mutant.** The scatter plot diagram shows the fluorescence intensity of CheO-sfGFP distributed along with the axe of 100 individual cells from the center (0) to the pole (1.0). The black line represents the average intensity of each measuring point. The histogram

shows the quantification of the CheO-sfGFP signal intensity at the pole (1.0) and the center (0) of each cell. Data presented as mean ± SEM. Scale bar, 5 μm.
(TIF)

**S9 Fig. Analysis of *C. jejuni fliM*<sup>L99F</sup> mutants in microaerobic condition. (A)** Soft agar motility assay of *C. jejuni* wild-type; Δ*fliMY* mutant; Δ*fliMY*+*fliMY* strain and Δ*fliMY* + *fliM*<sup>L99F</sup>*fliY* strain. **(B)** Soft agar motility assay of *C. jejuni* wild-type; Δ*fliMY* mutant; Δ*cheVA* mutant; Δ*fliMY* + *fliM*<sup>L99F</sup>*fliY* strain and Δ*cheVA*Δ*fliMY* + *fliM*<sup>L99F</sup>*fliY* strain. The Δ*motA* strain as a negative control.
(TIF)

**S10 Fig. Distribution of CheO homologs and chemoreceptors involved in energy taxis or with PAS domain in species of the *Campylobacterota* phylum.** Neighbor-joining phylogenetic tree of *Campylobacterota* was built from alignments of 16s rRNA. Species containing the F3 chemosensory class are highlighted in red.
(TIF)

**S1 Table. Strains and plasmids used in this study.**
(XLSX)

**S2 Table. Primers used in this study.**
(XLSX)

**S1 Movie. Swimming ability of Δ*cheO* mutant.** Δ*cheO* cell swimming was recorded by a Zeiss Axio Observer A1 microscope equipped with a 100× Plan lens and a Zeiss AxioCam 503 color CCD camera.
(MP4)

# Acknowledgments

We thank Dr. Mark Goulian from University of Pennsylvania for his helpful suggestions regarding fluorescence imaging of bacterial cells under microaerobic and aerobic switch.

# Author Contributions

**Conceptualization:** Beile Gao.

**Data curation:** Beile Gao.

**Formal analysis:** Ran Mo, Beile Gao.

**Funding acquisition:** Beile Gao.

**Investigation:** Ran Mo, Wenhui Ma, Weijie Zhou, Beile Gao.

**Resources:** Wenhui Ma, Weijie Zhou.

**Supervision:** Beile Gao.

**Validation:** Beile Gao.

**Writing – original draft:** Ran Mo.

**Writing – review & editing:** Beile Gao.

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
