## [Decision Letter · Decision Letter 0]

26 Aug 2022

Dear Dr. Gao,

Thank you very much for submitting your manuscript "Polar localization of CheO under hypoxia promotes Campylobacter jejuni chemotactic behavior within host" for consideration at PLOS Pathogens. As with all papers reviewed by the journal, your manuscript was reviewed by members of the editorial board and by several independent reviewers. In light of the reviews (below this email), we would like to invite the resubmission of a significantly-revised version that takes into account the reviewers' comments.

Three experts in the field of bacterial motility and chemotaxis reviewed this work. Overall, the Reviewers concluded that this work is interesting as it identifies a new type of chemotaxis protein in Campylobacter and related species that has the potential to sense or be affected by oxygen tension or redox status. However, two Reviewers believe that there are significant experiments and details lacking in the work that leave too much ambiguity to develop strong conclusions for how CheO functions in chemotaxis in C. jejuni. The work was also reviewed by an editor and the editor largely concurs with all of the Major Issues listed by Reviewers 2 and 3. The authors should seriously consider all of these comments in the Major Issues (along with many of the Minor Issues) to improve the work. Furthermore, an editor notes that significant details are lacking from the Materials and Methods, such as detailed protocol for how the co-immunoprecipitation studies were performed (how were the proteins mixed together - purified proteins, proteins in cell lysates, mixing and washing conditions etc) and other methods that would limit the ability of others to perform similar analyses. The authors should examine this section and provide additional details for this method and potentially others.

We cannot make any decision about publication until we have seen the revised manuscript and your response to the reviewers' comments. Your revised manuscript is also likely to be sent to reviewers for further evaluation.

Sincerely,

David R Hendrixson

Guest Editor

PLOS Pathogens

Nina Salama

Section Editor

PLOS Pathogens

Kasturi Haldar

Editor-in-Chief

PLOS Pathogens

orcid.org/0000-0001-5065-158X

Michael Malim

Editor-in-Chief

PLOS Pathogens

orcid.org/0000-0002-7699-2064

Three experts in the field of bacterial motility and chemotaxis reviewed this work. Overall, the Reviewers concluded that this work is interesting as it identifies a new type of chemotaxis protein in Campylobacter and related species that has the potential to sense or be affected by oxygen tension or redox status. However, two Reviewers believe that there are significant experiments and details lacking in the work that leave too much ambiguity to develop strong conclusions for how CheO functions in chemotaxis in C. jejuni. The work was also reviewed by an editor and the editor largely concurs with all of the Major Criticisms listed by Reviewers 2 and 3. The authors should seriously consider all of these comments to improve the work. Furthermore, an editor notes that significant details are lacking from the Materials and Methods, such as detailed protocol for how the co-immunoprecipitation studies were performed (how were the proteins mixed together - purified proteins, proteins in cell lysates, mixing and washing conditions etc) and other methods that would limit the ability of others to perform similar analyses. The authors should examine this section and provide additional details for this method and potentially others.

Reviewer's Responses to Questions

**Part I - Summary**

Reviewer #1: This is a very comprehensive, elegant and detailed study of a new chemotaxis regulator in campylobacters.

The manuscript is well written, and the study will greatly contribute the chemotaxis field in general.

There are a few thoughts for the authors to consider and some minor language corrections:

In your discussion:

Consider postulating that CheO is likely to associate with CheA within the chemosensory array in a manner similar to CheY and that is the reason that once the array formation is destabilised, it does not localise to the poles.

Line 361-364: Consider that vigorous motility does not really allow for adaptation, but rather allows the cell to sense and relocate to more favourable or preferred oxygen environment/level.

There are a few suggestions for the discussions within the minor comments below:

Reviewer #2: The manuscript describes a chemotaxis protein responsive to oxygen, named CheO. CheO homologs are conserved in Campylobacter, Helicobacter and Wolinella. The authors show that a functional CheOsgfp localizes at cell poles in a CheAVW-dependent manner, suggesting its localization depends on chemotaxis signaling array formation. CheO-sgfp polar localization is observed when cells are grown under microaerobic conditions but is lost under aerobic conditions. The contribution of CheO to chemotaxis is also increased under microarobic conditions compared to aerobic conditions. CheO appears to affect the swimming reversal, but not speed, and the authors provide evidence that it physically interacts with CheA, CheZ, FliM and FliY. The authors also show that CheO polar localization depends on the presence of energy taxis receptors (tlp6, tlp9aer1 and aer2). The findings are potentially significant and are certainly novel. However, some of the experiments are incomplete and some conclusions are preliminary with the data presented. The discussion is also highly speculative and lacks citation of relevant references. this reviewer suggests a few additional experiments and a more focused discussion and data presentation.

Reviewer #3: Mo and colleagues report the study of the newly identified CheO protein of C. jejuni. They present a solid characterization showing that cheO is important for chemotaxis particularly under microaerobic conditions. The protein localizes to the pole more under microaerobic conditions, possibly due to interactions with a subset of chemoreceptors. This part of the manuscript is interesting, and could be developed more. The authors show that CheO interacts with multiple chemotaxis and flagella proteins. These findings are well supported, but confusing because it seems surprising for a protein to have such wide interactions, and given that all are present in aerobic and microaerobic, not clear why CheO would lose the interactions and move off the pole. Overall, this work is interesting but would be more powerful if further developed.

**Part II – Major Issues: Key Experiments Required for Acceptance**

Reviewer #1: no major issues

Reviewer #2: 1. CheO is proposed to physically interact with CheA, CheZ, FliM and FliY and to localize to the cell poles in a CheAVW dependent manner, as expected as well as in presence of tlp6, tlp9aer1aer2. It is unclear how these multiple interactio0ns as well as an "energy taxis" receptors dependent localization and major role under microaerobic conditions may take place but a few experiments could clarify the findings. Does CheO interact with the energy taxis receptors? Does CheO-sGFP localization at the cel poles in aerobic versus microaerobic conditions depends on CheAVW as well as these energy taxis receptors? What is the expression patterns of the energy taxis receptors in aerobic versus microaerobic conditions?

2. The authors provide strong evidence that CheO interacts with FliM (ppi assays, suppressor analysis and role of cheO on swimming reversals). However, the discussion and most conclusions revolve around the link between cheO, microaerobiosis and energy taxis receptors. How does CheO-sgfp localize in mutants lacking FliY, FliM or relative to these proteins? Are all of these proteins expressed equally in aerobic versus microaerobic conditions?

3. Discussion : the last two paragraphs lack relevant references that include the published role of Tlp6 CZB in Helicobacter, the subcellular localization of TlpD in H. pylori that is relevant here and the aer1/aer2 and PAS domains of Tlp9 as well as what is known about energy taxis in these different species. Given the multiple interactions of CheO suggested here, additional information on the flagellar motor and chemotaxis in this species should be included. As is, the discussion is speculative and too removed from the existing literature.

Reviewer #3: Major comments

1. The expression of cheO under aerobic and microaerobic conditions. Fig 3E shows that the expression level of cheO gene remains unchanged in aerobic and microaerobic condition. What about the protein, which could be done by anti-GFP? This is important because if we look at the middle picture of panel A (Fig. 3), the total fluorescence intensity of cheO-sfGFP along the cell body is very different between aerobic and microaerobic conditions. The fluorescence intensity of CheO-sfGFP in microaerobic condition is much higher than that under aerobic condition. The expression level of gfp could affect GFP intracellular distribution.

2. The result of CheO relocalization is very interesting--but having the GFP fusion could be introducing artifacts. Could authors confirm this the localization of CheO under different conditions using immonofluorescence with untagged CheO, or at least differently tagged CheO?

3. The authors show that CheO is dispersed in the quadruple tlp9, aer1, aer2, tlp6 mutant, but not in any single chemoreceptor mutants. This work needs follow up. First, are CheA, CheW, CheV dispersed in these quadruple mutants, e.g. is it specific? Line 241-245 and Fig. S5, not all the single tlp/aer mutants are shown, e.g. no single tlp9, aer1 or aer2 deletion. Also, the mutant labeled by deltatlp9aer1aer2 is a bit unclear-- should it be labeled as deltatlp9 deltaaer1 deltaaer2.

4. It would be great to include double and triple tlp6, tlp9, aer1, aer2 mutants to really understand which receptor(s) are driving the localization.

5. What is the soft agar migration of the quadruple receptor mutant?

6. Were the protein-protein interactions done under aerobic or microaerobic conditions? Do any of them change?

7. Suppressors, Fig. 5. --The authors should show whether the FliM mutation is specific to cheO, or whether it causes increased migration of other chemotaxis mutants and WT.

**Part III – Minor Issues: Editorial and Data Presentation Modifications**

Reviewer #1: Minor corrections:

Line 155: This is not clear, and I think it meant to say that the mutation had reduced, rather than abolished the ability of bacteria to swarm on the soft agar plate. The mutation did not affect swimming…

Line 157: swarming?

Line 161, remove coma after serine

Line 166, effect ON chemotaxis

Line 206, please rephrase the subtitle, it is not clear, perhaps: “Tlps involved in energy…”

Lines 220-223, please rephrase for clarity.

Line 225, Line 350: swarming rather than spreading?

Line 231: not likely to be regulated…

Line 231: residues

Line 237: appears less likely to be able…

Line 246-247: in the following statement, please clarify whether Tlp6 or Aer1/2 are composed (actually: contain) a PAS domain.

Tlp9 interacts with sensor proteins Aer1/Aer2 composed of the PAS domain alone and Tlp6 has a CZB domain

Line148: throughout, rather than “all over”

Line 249-251: Please consider that it is not the coordinate presence of Tlps that co-ordinately determine that recruitment of CheO, but the mutation you created had destabilised the chemosensory array formation leading to inability of CheO to integrate/associate. I think this is more likely, and in any case should be considered, both here and in discussion.

Lines 285-296. C. jejuni NCTC 11168GC is a known oxygen adapted variant. It had been previously sequenced and, as compared to the 11168-O, it had a number of mutations, thought to allow it to be oxygen tolerant. Are any of those mutations correlate with your revertant’ s point mutations? A nice point to include in your discussion, one way or the other.

Line 351: dispersed throughout the cell.

Reviewer #2: 1. chemoarray is not the term used: it should be chemosensory arrays or chemotaxis signaling arrays.

2. lines 264-265: the REC domain of CheA is NOT a CheY but a response regulator domain. it should NOT be labeled as CheAY- this is a hybrid kinase CheA with a C0-terminal REC domain which function has not been established in Campylobacter.

3. lines 271-273: a reference is needed for this statement.

4. lines 345-346: the authors note some phosphorylation assays but these are not described in the methods or presented in the data. A role for phosphorylation may also0 not be the most relevant here.

5.line 246: TlpD in H. pylori senses oxidative stress, not oxygen.

Reviewer #3: Minor comments

8. Abstract--include the cheO gene number to make it easier to compare to other work (CJJ81176_1265)

9. Please include information about the predicted cheO genomic location, operon, and the 2º structure of the protein. Is it predicted to be cytoplasmic?

10. Line 166-170, single-cell tracking was used to test the role of CheO in chemotaxis (Figs. 1F and G). Is the swimming behavior tracked without adding chemotaxis ligands? The conditions and cognate ligands used in this assay should be indicated.

11. Figure 1C lacks the statistical analysis.The caption about Fig 1F and G is incomplete.

12. Line 187-189 (Fig. 2A). Is the fluorescence observation done under microaerobic conditions? The description is not clear.

13. Figure 3A. The backgrounds of fluorescence pictures is abnormal. Did authors adjust the image contrast or lightness a lot to make the backgrounds vague? Please make this clear in the figure legend and methods.

14. Figure 3C and D need statistical analysis.

15. Line 246-247 ‘Tlp9 interacts with sensor proteins Aer1/Aer2…’ needs citation.

16. Figure 4 shows that CheO interacts with both the HPT and linker region of CheA. Do authors have any idea about how does that work? For this and the other interactions, it might be useful to determine the cognate interacting region of CheO with other proteins/domains?

17. Fig. 4 might work better moved to come before Fig. 3.

18. Line 292: The authors should make it more clear that the G297T refers to nucleotides (not AA, as I initially thought).

19. The discussion part is pretty simplified. It looks like that CheO can interact with many proteins including CheA, CheZ, FliM, and FliY. What do the authors believe is the reason that CheO can interact so many proteins with different functions and domains?

20. Methods: There are multiple plate names that are not clearly defined, e.g. karmali agar, and sometimes use of TSA, Brucella Broth, Blood (not clearly defined). Please check the media used and make sure each is clearly described.

21. Figure S3 caption about ‘strains expressing cheO-sfGfp or sfGFP-cheY’ is unclear, wild-type strain or cognate mutants?

PLOS authors have the option to publish the peer review history of their article (what does this mean?). If published, this will include your full peer review and any attached files.

Reviewer #1: No

Reviewer #2: No

Reviewer #3: No
---

## [Editor Report · Decision Letter 1]

27 Oct 2022

Dear Dr. Gao,

We are pleased to inform you that your manuscript 'Polar localization of CheO under hypoxia promotes Campylobacter jejuni chemotactic behavior within host' has been provisionally accepted for publication in PLOS Pathogens.

Best regards,

David R Hendrixson

Guest Editor

PLOS Pathogens

Nina Salama

Section Editor

PLOS Pathogens

Kasturi Haldar

Editor-in-Chief

PLOS Pathogens

orcid.org/0000-0001-5065-158X

Michael Malim

Editor-in-Chief

PLOS Pathogens

orcid.org/0000-0002-7699-2064

Thank you for thoughtfully considering and addressing all of the Reviewers' comments. The manuscript is improved and more insight has been provided for the role of CheO in hypoxia and cheomotaxis in Campylobacter jejuni.
---

## [Editor Report · Acceptance letter]

31 Oct 2022

Dear Dr. Gao,

We are delighted to inform you that your manuscript, "Polar localization of CheO under hypoxia promotes Campylobacter jejuni chemotactic behavior within host," has been formally accepted for publication in PLOS Pathogens.

Best regards,

Kasturi Haldar

Editor-in-Chief

PLOS Pathogens

orcid.org/0000-0001-5065-158X

Michael Malim

Editor-in-Chief

PLOS Pathogens

orcid.org/0000-0002-7699-2064